

# CD4 cell count and CD4/CD8 ratio increase during rituximab maintenance in granulomatosis with polyangiitis patients

Emilio Besada[1] and Johannes C. Nossent[2,3]

[1] Bone and Joint Research Group, Institute of Clinical Medicine, Faculty of Health Sciences, UiT The Arctic University of Norway, Tromsø, Norway
[2] School of Medicine & Pharmacology QEII Medical Centre Unit, University of Western Australia, Australia
[3] Rheumatology, Sir Charles Gairdner Hospital, Nedlands, WA, Australia

Corresponding author
Emilio Besada, emilio.besada@uit.no

## ABSTRACT

**Introduction:** Rituximab (RTX) is a B cell-depleting agent approved for the treatment of granulomatosis with polyangiitis (GPA). RTX reduces antibody producing precursor plasma cells and inhibits B and T cells interaction. Infections related to T cell immunodeficiency are not infrequent during RTX treatment. Our study investigated CD4 cell count and CD4/CD8 ratio in GPA patients during the first two years of long-term RTX treatment.

**Methods:** A single centre cohort study of 35 patients who received median total cumulative dose of cyclophosphamide (CYC) of 15 g and were treated with RTX 2 g followed by retreatment with either 2 g once annually or 1 g biannually. Serum levels of total immunoglobulin (Ig) and lymphocytes subsets were recorded at RTX initiation and at 3, 6, 12, 18 and 24 months. Low CD4 count and inverted CD4/CD8 ratio were defined as CD4 $< 0.3 \times 10^9$/l and ratio $< 1$.

**Results:** The CD4 cell count and CD4/CD8 ratio decreased slightly following the initial RTX treatment and then increased gradually during maintenance treatment. While the proportion of patients with low CD4 cell count decreased from 43% at baseline to 18% at 24 months, the ratio remained inverted in 40%. Oral daily prednisolone dose at baseline, CYC exposure and the maintenance regimen did not influence the CD4 cell count and ratio. Being older (p = 0.012) and having a higher CRP (p = 0.044) and ESR (p = 0.024) at baseline significantly increased the risk of inverted CD4/CD8 ratio at 24 months. Inverted ratio at baseline associated with lower total Ig levels during the study.

**Conclusions:** Overall, the CD4 and CD4/CD8 ratio increased during maintenance RTX therapy in GPA with no discernible impact of other immunosuppressive therapy. However the increase in CD4 was not followed by an increase in the CD4/CD8 ratio, especially in older patients. Inverted CD4/CD8 ratio associated with lower Ig levels, suggesting a more profound B cell depleting effect of RTX with a relative increase in CD8+ lymphocytes.

## INTRODUCTION

Granulomatosis with polyangiitis (GPA) is an antineutrophil cytoplasmic antibody (ANCA)-associated vasculitis (AAV) resulting in a necrotizing small to medium vessels vasculitis and a necrotizing granulomatous inflammation involving predominantly the upper and lower respiratory tract and the kidneys. GPA is the result of a complex interplay between the humoral and cellular immunity involving proteinase 3-ANCA (PR3-ANCA), neutrophils, endothelial cells, B and T cells (*Kallenberg, 2011*).

T cells are important in GPA as persistent activation of T cells through aberrant expression of costimulatory molecules favours the expansion of effector memory T cells and the formation of granuloma (*Wilde et al., 2010*). Changes in T cells in GPA patients occur during remission when circulating memory T cells are increased and naive T CD4+ cells are decreased (*Abdulahad et al., 2006*). Also, T cell-targeted therapies such as abatacept, alemtuzumab and gusperimus are alternative treatments to B cell depletion with rituximab (RTX) in GPA (*Furuta & Jayne, 2014*).

However, a subgroup of GPA patients has low CD4 cell count and inverted CD4/CD8 ratio irrespective of disease activity and the use of immunosuppressive drugs, possibly due to the recruitment of T cells into the inflamed tissue (*Berden et al., 2009*). It is not clear whether it is the decrease of CD4 cell count (*Marinaki et al., 2006*), the expansion of CD8 cell count or a combination of both (*Ikeda et al., 1992*; *Iking-Konert et al., 2008*) that is responsible for the inversion of the ratio.

RTX is a chimeric human-mouse monoclonal antibody directed against CD20 that induces rapid and sustained depletion of premature and mature B cells through antibody-dependent, complement-mediated cellular cytotoxicity and apoptosis (*Leandro & de la Torre, 2009*). RTX reduces auto-antibody producing precursor plasma cells, inhibits B cell interaction with auto reactive T cells and decreases the level of soluble factors secreted by B cells (*Leandro & de la Torre, 2009*). RTX is approved for the treatment of rheumatoid arthritis (RA) and AAV and is used off label in a large number of autoimmune conditions (*Edwards & Cambridge, 2006*). In AAV, RTX is used to induce (*Jones et al., 2010*; *Stone et al., 2010*) and to maintain (*Guillevin et al., 2014*; *Smith et al., 2012*; *Cartin-Ceba et al., 2012*; *Besada, Koldingsnes & Nossent, 2013*) remission through iterative infusions. Relevant side effects of RTX include late onset neutropenia (*Besada, Koldingsnes & Nossent, 2012*), hypogammaglobulinemia (*Makatsori et al., 2014*; *Besada, Koldingsnes & Nossent, 2014*) and an increased risk of infections (*Besada, Koldingsnes & Nossent, 2013*; *Ram et al., 2009*; *Sailler et al., 2008*; *Gottenberg et al., 2010*). In two studies, 27–44% of all severe infections during RTX treatment were either viral or fungal (*Besada, Koldingsnes & Nossent, 2013*; *Makatsori et al., 2014*), possibly related to T cell immunodeficiency.

Our study investigated the course of CD4 cell count, CD4/CD8 ratio and serum levels of total immunoglobulin (Ig) in GPA patients receiving long-term RTX treatment.

## METHODS

The Vasculitis Registry in Northern Norway is an observational prospective registry collecting data on disease presentation and course from patients with an established

diagnosis of primary vasculitis. All patients gave informed written consent at inclusion in accordance with the declaration of Helsinki.

A total of 35 patients from the Vasculitis Registry in Northern Norway with an established diagnosis of GPA were included in the study. The study did not require formal ethical approval and conformed to the standards currently applied in Norway. Patients' characteristics at baseline have been previously described (*Besada, Koldingsnes & Nossent, 2013*).

At RTX initiation, the patients (median age of 50 (14–79), 54% males) had median disease duration of 55 months (1–270). However, 86% of the patients were ANCA positive; all except one were PR3-ANCA positive. Renal, lung and orbital or/and subglottic involvement was present in 60, 63 and 57% of the patients. Median Birmingham Vasculitis Activity Score (BVAS) was 9 (0–22) at baseline. The main indication for RTX was disease relapse (80%), new disease onset (17%) and maintenance therapy (3%). Patients had received a median cumulative cyclophosphamide (CYC) dose of 14 g (0–250) prior to RTX and all had normal total Ig levels (> 6 g/l) prior to RTX.

Patients received 2 g RTX at induction (1 g twice in a fortnight) with co-administration of methylprednisolone 125 mg, paracetamol 1,000 mg and either cetirizine 10 mg or polaramine 4 mg. RTX was usually combined with a median oral daily prednisolone dose (ODPD) of 20 mg (0–60) and an immunosuppressive drug in 91% of the patients. Overall, patients received a median cumulative RTX dose of 4 g (2–6) during the first 24 months following its initiation. Respectively 49 and 40% received long-term maintenance with the 2 g annually regimen (1 g twice in a fortnight per year) or the 1 g biannually regimen (1 g every six months), while 11% received only induction.

Clinical parameters such as gender, age, erythrocyte sedimentation rate (ESR, nr: < 20 in men and < 28 mm/h in women), C-reactive protein (CRP, nr < 5 mg/l), creatinine (nr: 60–105 in men and 45–90 $\mu$mol/l in women), ANCA titers (nr < 10 IU/ml), the ODPD and the cumulative dose of CYC were recorded at baseline. Serum levels of total Ig and lymphocytes subsets were measured at RTX initiation and at 3, 6, 12, 18 and 24 months in 35, 24, 31, 34, 31 and 34 patients, respectively. CD4 and CD8 cell counts were determined by flow cytometry in blood specimens. Normal adult CD4 and CD8 cell counts for our laboratory ranged from 0.3–1.4 and 0.2–0.9 $\times 10^9$/l, respectively. Low CD4 cell count was defined as CD4 $< 0.3 \times 10^9$/l and inverted CD4/CD8 ratio when ratio < 1. Hypogammaglobulinemia was defined as serum total Ig < 6 g/l. Severe infections were defined as infections necessitating intravenous antibiotic treatment and/or hospitalisation.

Data were analysed with SPSS version 20.0 (SPSS Ltd, Chicago, IL, USA). The results are expressed in percentage for categorical variable and in median (range) for continuous variables, unless specified otherwise. Fisher's exact test, Wilcoxon signed rank test and Man-Whitney U test were used as appropriate. Significant predictors of inversion of the CD4/CD8 ratio at 24 months determined during univariable analysis were entered in a multivariable binary logistic regression model with backward selection ($p < 0.05$ to enter and $p < 0.10$ to stay). Missing data were excluded from the statistical analyses. p-values < 0.05 were considered significant.

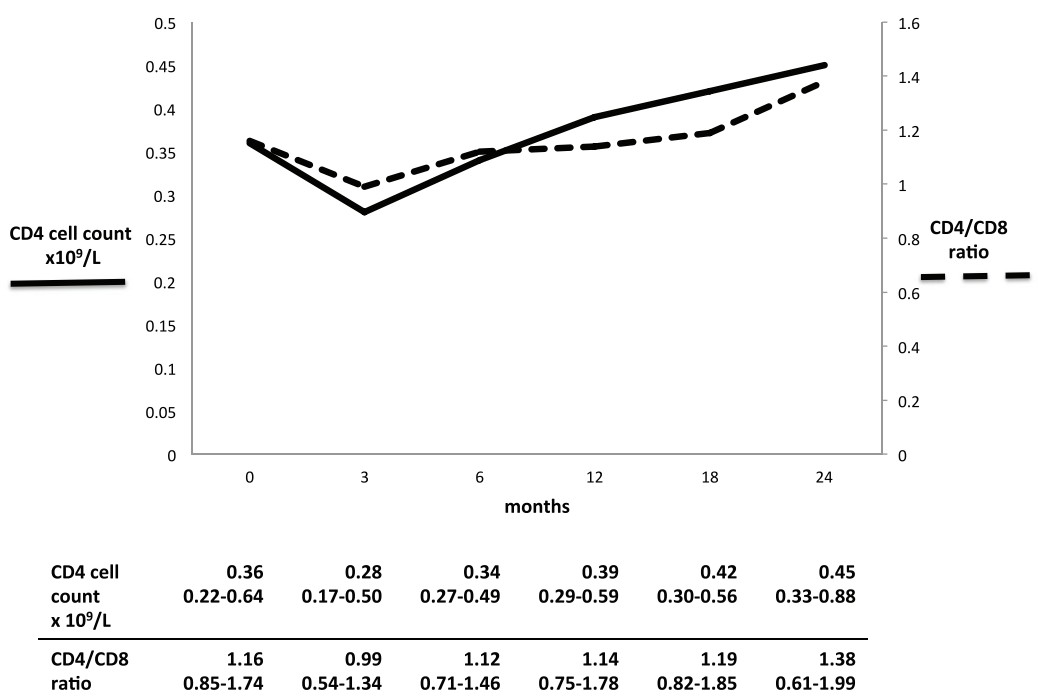

| | | | | | | |
|---|---|---|---|---|---|---|
| CD4 cell count x 10⁹/L | 0.36 0.22-0.64 | 0.28 0.17-0.50 | 0.34 0.27-0.49 | 0.39 0.29-0.59 | 0.42 0.30-0.56 | 0.45 0.33-0.88 |
| CD4/CD8 ratio | 1.16 0.85-1.74 | 0.99 0.54-1.34 | 1.12 0.71-1.46 | 1.14 0.75-1.78 | 1.19 0.82-1.85 | 1.38 0.61-1.99 |

**Figure 1  CD4 cell count and CD4/CD8 ratio in GPA patients during long-term RTX treatment.** Full line: low CD4 cell count. Dashed line: inverted CD4/CD8 ratio. Table results are expressed in median and interquartile range.

# RESULTS

## CD4 cell count and ratio

Both the CD4 cell count and the CD4/CD8 ratio decreased initially after RTX administration, and thereafter increased gradually (Fig. 1). The CD4 cell count seemed to decrease between baseline and three months ($0.36-0.28 \times 10^9$/l, p = 0.166), while the ratio decreased significantly from 1.16 to 0.99 (p = 0.011). At 24 months, CD4 cell count and ratio had both significantly increased from baseline: $0.45 \times 10^9$/l (p = 0.039) and 1.38 (p = 0.031), respectively.

However, while the proportion of patients with a low CD4 cell count decreased from 43% at baseline to 18% at 24 months, the proportion of patients with inverted ratio remained stable around 40% throughout the study period (Fig. 2).

## Baseline clinical profile influence on CD4 cell count and ratio
### CD4 cell count and ratio at baseline

Patients with low CD4 cell count and inverted ratio at baseline had a tendency to be older (p = 0.069 and p = 0.057, respectively) (Tables 1 and 2). Only patients with low CD4 cell count seemed to have a higher cumulative dose of CYC compared with patients with a normal baseline CD4 cell count (p = 0.071) (Table 2).

There were no difference in organ involvements, disease duration prior to RTX, BVAS, CRP, ESR, creatinine and ANCA titers at RTX initiation between patients with normal and low CD4 cell count at baseline (Table 1) and patients with normal and inverted ratio (Table 2).

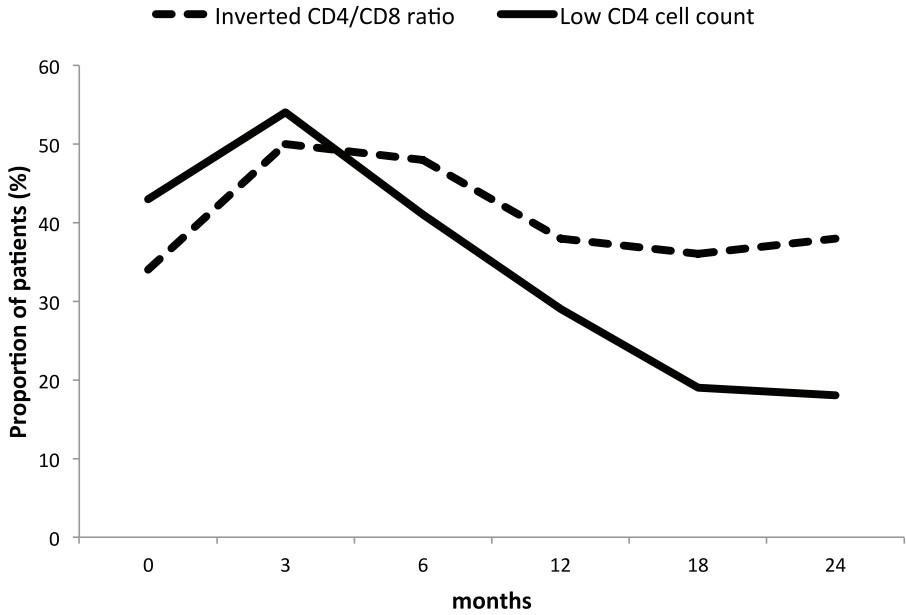

**Figure 2 Proportion of GPA patients with low CD4 cell count and inverted ratio during long-term rituximab.** Full line: low CD4 cell count. Dashed line: inverted CD4/CD8 ratio. The proportion of patients is expressed in percentage.

### CD4/CD8 ratio at 24 months

Patients who had an inverted CD4/CD8 ratio at 24 months were older (60 vs. 45 years, p = 0.003), had higher ESR (43 vs. 12 mm/h, p = 0.009) and creatinine (82 vs. 67 μmol/l, p = 0.018) at baseline. Although not significant, they also had a tendency to higher BVAS (11 vs. 8, p = 0.096), CRP (24 vs. 6 mg/l, p = 0.076) and ANCA titers (16 vs. 5 IU/ml, p = 0.089). Being older and having higher CRP and ESR at baseline significantly increased the risk of inverted CD4/CD8 ratio at 24 months during univariable analysis (Table 3). Age was the most important predictor for inversion of the CD4/CD8 ratio at 24 months during the multivariable analysis (Table 3). Being 10 years older at RTX initiation increased the risk by 2.5 times.

### Serum immunoglobulin levels

Serum total Ig levels in both patients with low CD4 cell count and inverted ratio were lower at all time points during RTX maintenance (Tables 1 and 2). This was only significant in patients with inverted ratio compared with patients with normal ratio at baseline (Table 2). The different RTX maintenance regimens did not influence CD4 cell count, ratio and total Ig levels in the first two years (Table 4).

### Severe infections in the first 24 months after RTX initiation

Two patients (5.7%) had severe infections during the study period three and four months after RTX initiation. One patient had sinusitis secondary to Pseudomonas aeruginosa and the other had Pneumocystis jiroveci pneumonia. They were men, aged 62 and 79 years old, who had received 250 and 25 g of CYC. They had low B cell (0.04 and 0.06 × $10^9$/l) and CD4 cell counts (0.17 and 0.26 × $10^9$/l), inverted ratio (0.35 and 0.43), but had

**Table 1 Characteristics of GPA patients with low CD4 cell count at baseline against the rest of the cohort.** Results are expressed in number (percentage) for categorical variables and median (range) for continuous variables. Differences were determined respectively by Fisher's exact test and Mann Whitney U test. Serum total immunoglobulin level and CD4 cell count expressed respectively in g/L and $\times 10^9$/L. Significant results are highlighted in bold.

| | Low CD4 count 15 patients | Normal CD4 count 20 patients | p-value |
|---|---|---|---|
| Male | 9 (40%) | 10 (50%) | 0.784 |
| Age y | 58 (19–79) | 47 (14–67) | 0.069 |
| Disease duration prior RTX mo | 93 (2–270) | 45 (1–198) | 0.254 |
| BVAS at baseline | 11 (4–21) | 8.5 (0–21) | 0.657 |
| ESR mm/t | 15 (3–118) | 17 (7–100) | 0.400 |
| CRP mg/L | 6 (4–135) | 7.5 (4–127) | 0.908 |
| Creatinine μmol/L | 75 (46–244) | 77 (43–819) | 0.934 |
| ANCA titers IU/L | 7 (0–531) | 8 (1–109) | 0.780 |
| Kidney involvement | 8 (53%) | 13 (65%) | 0.511 |
| Lung involvement | 10 (67%) | 12 (60%) | 0.737 |
| Orbital-subglottic involvement | 10 (67%) | 10 (50%) | 0.492 |
| ODPD mg | 25 (3–50) | 20 (0–60) | 0.458 |
| Total CYC dose g | 25 (0–250) | 13 (0–68) | 0.071 |
| Total RTX dose g | 4.0 (2–5) | 4.0 (2–6) | 0.780 |
| 1 g biannually RTX regimen | 6 (40%) | 8 (40%) | 1.00 |
| Baseline | | | |
|     Total Ig | 10.0 (8.2–18) | 11.8 (6.7–21) | 0.283 |
|     HypoG | 0 | 0 | NA |
|     CD4 | **0.21 (0.06–0.29)** | **0.61 (0.31–1.4)** | **<0.001** |
|     Ratio | **0.94 (0.27–2.3)** | **1.51 (0.55–3.0)** | **<0.001** |
| At three months | | | |
|     Total Ig | 7.3 (4.8–15) | 8.7 (6.6–12) | 0.247 |
|     HypoG | 3 (25%) | 0 | 0.096 |
|     CD4 | 0.17 (0.09–1.4) | 0.32 (0.17–0.89) | 0.064 |
|     Ratio | 0.81 (0.34–1.5) | 1.12 (0.26–2.4) | 0.096 |
| At six months | | | |
|     Total Ig | 7.4 (4.5–16) | 8.4 (4.1–12) | 0.238 |
|     HypoG | 3 (23%) | 2 (11%) | 0.374 |
|     CD4 | **0.27 (0.16–0.57)** | **0.43 (0.24–0.93)** | **< 0.001** |
|     Ratio | **0.93 (0.38–1.3)** | **1.25 (0.42–4.5)** | **0.014** |
| At 12 months | | | |
|     Total Ig | 7.1 (4.9–15) | 8.5 (5.3–15) | 0.158 |
|     HypoG | 1 (7.1%) | 2 (10%) | 1.00 |
|     CD4 | **0.29 (0.17–0.64)** | **0.49 (0.19–1.4)** | **0.002** |
|     Ratio | **0.85 (0.33–1.8)** | **1.4 (0.51–4.8)** | **0.010** |

|  | Low CD4 count 15 patients | Normal CD4 count 20 patients | p-value |
|---|---|---|---|
| **At 18 months** |  |  |  |
| Total Ig | 6.7 (4.7–17) | 7.9 (4.9–14) | 0.297 |
| HypoG | 3 (21%) | 1 (5.9%) | 0.304 |
| CD4 | **0.35 (0.13–1.3)** | **0.49 (0.21–1.1)** | **0.042** |
| Ratio | **0.92 (0.40–1.9)** | **1.5 (0.53–6.0)** | **0.022** |
| **At 24 months** |  |  |  |
| Total Ig | 6.7 (3.0–14) | 7.5 (4.9–14) | 0.179 |
| HypoG | 5 (33%) | 3 (16%) | 0.417 |
| CD4 | **0.35 (0.06–1.0)** | **0.70 (0.08–1.3)** | **0.021** |
| Ratio | 0.84 (0.34–2.9) | 1.40 (0.36–5.4) | 0.120 |

**Note:**
ANCA, antineutrophil cytoplasmic antibodies; BVAS, Birmingham vasculitis activity score; CD, cluster of differentiation; CRP, C-reactive protein; CYC, cyclophosphamide; ESR, erythrocyte sedimentation rate; HypoG, hypogammaglobulinemia; Ig, immunoglobulins; mo, months; NA, not available; ODPD, oral daily prednisolone dose; RTX, rituximab.

normal serum levels of total Ig (11.2 and 12.6 g/l) at baseline. At the time of infection, CD4 cell count remained low (0.16 and $0.17 \times 10^9$/l) and total Ig had declined by to 4.8 and 7.5 g/l, respectively.

## CD4 cell count and ratio in GPA patients who did not receive RTX maintenance

Four GPA patients (two men and two women) with a median age of 65 (14–79) years who had received a cumulative dose of 3 g (0–25) g of CYC were only administered RTX at induction and did not receive RTX maintenance.

The CD4 cell count and ratio decreased from 0.54 (0.26–0.98) at baseline to 0.36 (0.34–0.63) $\times 10^9$/l at 24 months. The CD4/CD8 ratio decreased from baseline and became inverted at 24 months: from 1.4 (0.43–2.2) to 0.61 (0.45–2.0).

## DISCUSSION

In GPA patients on long term RTX treatment, overall CD4 cell count and CD4/CD8 ratio initially decreased in the first three months and thereafter gradually increased independent of the ODPD at baseline, the CYC cumulative dose and the maintenance regimen. However in older patients, the inverted ratio at baseline remained unchanged after two years and was associated with lower levels of total Ig.

### Early effect of rituximab on CD4 cell count

Prior to RTX treatment, 34 and 43% of GPA patients had respectively inverted CD4/CD8 ratio and decreased absolute numbers of CD4 cells. Age and the cumulative CYC dose were the most closely associated parameters, although these were not found statistically significant in this small cohort.

During RTX treatment, the early decrease in CD4 cell count seemed dependent of the baseline CD4 count. The higher the CD4 cell count was at baseline, the more CD4 cell

**Table 2 Characteristics of GPA patients with inverted ratio at baseline against the rest of the cohort.**
Results are expressed in number (percentage) for categorical variables and median (range) for continuous variables. Differences were determined respectively by Fisher's exact test and Mann Whitney U test. Serum total immunoglobulin level and CD4 cell count expressed respectively in g/l and $\times 10^9$/l. Significant results are highlighted in bold.

| | Inverted ratio 12 patients | Normal ratio 23 patients | p-value |
|---|---|---|---|
| Male | 6 (50%) | 13 (57%) | 0.736 |
| Age years | 58 (37–79) | 48 (14–75) | 0.057 |
| Disease duration prior RTX mo | 116 (2–270) | 47 (1–198) | 0.263 |
| BVAS at baseline | 10.5 (4–21) | 8.0 (0–21) | 0.482 |
| ESR mm/t | 29 (3–118) | 15 (3–100) | 0.548 |
| CRP mg/l | 20 (4–135) | 6 (4–115) | 0.310 |
| Creatinine μmol/l | 76 (58–244) | 75 (43–819) | 0.694 |
| ANCA titers IU/ml | 6 (0–531) | 8 (1–213) | 0.878 |
| Kidney involvement | 8 (67%) | 13 (57%) | 0.721 |
| Lung involvement | 8 (67%) | 14 (61%) | 1.00 |
| Orbital-subglottic involvement | 10 (83%) | 15 (65%) | 0.282 |
| ODPD mg | 23 (3–50) | 23 (0–60) | 0.719 |
| Total CYC dose g | 25 (0–250) | 13 (0–79) | 0.363 |
| Total RTX dose g | 4 (2–5) | 4 (2–6) | 0.263 |
| 1 g biannually RTX regimen | 5 (42%) | 9 (39%) | 1.00 |
| Baseline | | | |
|     Total Ig | 11.3 (8.3–18) | 11.3 (6.7–21) | 0.461 |
|     HypoG | 0 | 0 | NA |
|     CD4 | **0.22 (0.06–0.79)** | **0.49 (0.15–1.4)** | **0.005** |
|     Ratio | **0.61 (0.27–0.97)** | **1.42 (1.0–3.0)** | **< 0.001** |
| At three months | | | |
|     Total Ig | 7.1 (4.8–15) | 8.7 (5.0–12) | 0.238 |
|     HypoG | 1 (13%) | 2 (12%) | 1.00 |
|     CD4 | 0.20 (0.09–0.70) | 0.29 (0.13–1.4) | 0.264 |
|     Ratio | **0.46 (0.34–1.1)** | **1.12 (0.26–2.4)** | **0.002** |
| At six months | | | |
|     Total Ig | **6.7 (4.1–16)** | **9.1 (5.0–12)** | **0.017** |
|     HypoG | 3 (27%) | 2 (10%) | 0.310 |
|     CD4 | **0.27 (0.16–0.42)** | **0.43 (0.18–0.93)** | **< 0.001** |
|     Ratio | **0.55 (0.38–1.1)** | **1.25 (0.65–4.5)** | **< 0.001** |
| At 12 months | | | |
|     Total Ig | **6.6 (5.3–15)** | **9.2 (4.9–15)** | **0.021** |
|     HypoG | 1 (8.3%) | 2 (9.1%) | 1.00 |
|     CD4 | **0.28 (0.19–0.39)** | **0.50 (0.17–1.4)** | **< 0.001** |
|     Ratio | **0.62 (0.33–1.3)** | **1.47 (0.77–4.8)** | **< 0.001** |

| Table 2 (continued). | Inverted ratio 12 patients | Normal ratio 23 patients | p-value |
|---|---|---|---|
| **At 18 months** | | | |
| Total Ig | **6.5 (4.7–17)** | **8.6 (5.5–13)** | **0.012** |
| HypoG | 3 (30%) | 1 (4.7%) | 0.087 |
| CD4 | **0.34 (0.15–0.48)** | **0.48 (0.13–1.3)** | **0.026** |
| Ratio | **0.53 (0.40–1.0)** | **1.63 (0.60–6.0)** | **< 0.001** |
| **At 24 months** | | | |
| Total Ig | 6.2 (3.0–14) | 7.9 (4.0–14) | 0.063 |
| HypoG | 5 (42%) | 3 (14%) | 0.098 |
| CD4 | **0.34 (0.06–1.02)** | **0.69 (0.13–1.3)** | **0.007** |
| Ratio | **0.56 (0.34–1.5)** | **1.64 (0.60–5.4)** | **< 0.001** |

**Note:**

ANCA, antineutrophil cytoplasmic antibodies; BVAS, Birmingham vasculitis activity score; CD, cluster of differentiation; CRP, C-reactive protein; CYC, cyclophosphamide; ESR, erythrocyte sedimentation rate; HypoG, hypogammaglobulinemia; Ig, immunoglobulins; mo, months; NA, not available; ODPD, oral daily prednisolone dose; RTX, rituximab.

**Table 3 Odds ratio for inverted CD4/CD8 ratio at 24 months in GPA patients receiving rituximab maintenance.** All values were determined at rituximab initiation and were analysed with an unadjusted and multivariable logistic regression models with backward Wald selection (removal if p < 0.10). Significant results are highlighted in bold.

| | Unadjusted analysis | | | Multivariable analysis | | |
|---|---|---|---|---|---|---|
| | OR | 95% CI | p-value | OR | 95% CI | p-value |
| Men | 1.06 | 0.27–4.24 | 0.934 | | | |
| **Age (y)** | **1.10** | **1.02–1.18** | **0.012** | **1.09** | **1.01–1.18** | **0.027** |
| Disease duration (mo) | 1.01 | 0.99–1.02 | 0.075 | 1.01 | 0.99–1.03 | 0.089 |
| BVAS | 1.13 | 0.96–1.32 | 0.135 | | | |
| **ESR (mm/h)** | **1.03** | **1.00–1.05** | **0.024** | 1.03 | 0.99–1.06 | 0.098 |
| **CRP (mg/l)** | **1.02** | **1.00–1.04** | **0.044** | | | 0.705 |
| Creatinine ($\mu$mol/l) | 1.02 | 0.99–1.05 | 0.081 | | | 0.164 |
| ANCA titers (IU/ml) | 1.03 | 0.98–1.08 | 0.184 | | | |
| ODPD (mg/d) | 1.04 | 0.98–1.09 | 0.138 | | | |
| CYC cumulative dose (g) | 1.01 | 0.99–1.03 | 0.388 | | | |
| IgG (g/l) | 1.13 | 0.84–1.52 | 0.428 | | | |
| IgA (g/l) | 1.20 | 0.50–2.93 | 0.682 | | | |
| IgM (g/l) | 2.21 | 0.70–7.00 | 0.177 | | | |
| Total Ig (g/l) | 1.11 | 0.89–1.37 | 0.357 | | | |

**Note:**

ANCA, antineutrophil cytoplasmic antibodies; BVAS, Birmingham vasculitis activity score; CI, confidence interval; CRP, C-reactive protein; CYC, cyclophosphamide; ESR, erythrocyte sedimentation rate; Ig, immunoglobulins; mo, months; ODPD, oral daily prednisolone dose; OR, odds ratio; y, years.

count declined at three months. GPA patients with normal CD4 cell count (mean $0.67 \times 10^9$/L at baseline) had a 40% decrease at three months while GPA patients with low CD4 cell count (mean $0.19 \times 10^9$/L at baseline) had an 89% increase.

**Table 4 Characteristics of the GPA patients receiving RTX 1 g biannually as maintenance treatment compared with the rest of the cohort.** Differences between categorical and continuous variables are determined respectively by Fisher's exact test and Mann Whitney U test. Serum total immunoglobulin level and CD4 cell count expressed respectively in g/L and $\times 10^9$/L.

| | 1 g biannually regimen 14 patients | Other regimens 21 patients | p-value |
|---|---|---|---|
| Male | 9 (64%) | 10 (48%) | 0.491 |
| Age y | 50 | 47 | 0.702 |
| Total CYC dose g | 43 | 25 | 0.274 |
| Total RTX dose g | 4.9 | 3.7 | < 0.001 |
| Baseline | | | |
|     Total Ig | 12.2 | 10.9 | 0.342 |
|     HypoG | 0 | 0 | NA |
|     CD4 | 0.39 | 0.51 | 0.474 |
|     Ratio | 1.31 | 1.37 | 0.881 |
| At three months | | | |
|     Total Ig | 8.4 | 8.6 | 0.426 |
|     HypoG | 3 (25%) | 0 | 0.096 |
|     CD4 | 0.25 | 0.46 | 0.194 |
|     Ratio | 0.80 | 1.11 | 0.123 |
| At six months | | | |
|     Total Ig | 8.8 | 7.9 | 0.600 |
|     HypoG | 2 (15%) | 3 (16%) | 1.00 |
|     CD4 | 0.41 | 0.41 | 0.667 |
|     Ratio | 1.51 | 1.20 | 0.421 |
| At 12 months | | | |
|     Total Ig | 8.5 | 8.9 | 0.594 |
|     HypoG | 1 (7.7%) | 2 (9.5%) | 1.00 |
|     CD4 | 0.45 | 0.49 | 0.600 |
|     Ratio | 1.72 | 1.32 | 0.834 |
| At 18 months | | | |
|     Total Ig | 8.3 | 8.4 | 0.666 |
|     HypoG | 4 (31%) | 0 | 0.023 |
|     CD4 | 0.37 | 0.57 | 0.059 |
|     Ratio | 1.78 | 1.41 | 0.953 |
| At 24 months | | | |
|     Total Ig | 7.5 | 8.2 | 0.650 |
|     HypoG | 3 (21%) | 5 (25%) | 1.00 |
|     CD4 | 0.52 | 0.61 | 0.522 |
|     Ratio | 1.91 | 1.42 | 0.823 |

**Note:**
CD, cluster of differentiation; CYC, cyclophosphamide; HypoG, hypogammaglobulinemia; Ig, immunoglobulins; RTX, rituximab.

The same pattern was observed in RA (*Mélet et al., 2013*; *Thurlings et al., 2008*; *Feuchtenberger et al., 2008*) and in systemic lupus erythematosus (SLE) patients (*Sfikakis et al., 2005*), but not in renal transplantation (*Kamburova et al., 2014*).

In RA, CD4 cell count decreased by 37% at three months from a mean of $1.25 \times 10^9$/l at baseline (*Mélet et al., 2013*), remained unchanged at four months from a mean of $0.93 \times 10^9$/l at baseline (*Thurlings et al., 2008*) and increased by 14% from a mean of $0.63 \times 10^9$/l at baseline during the first three months (*Feuchtenberger et al., 2008*). SLE patients had a 46% increase in CD4 cell count one month after RTX initiation from a mean of $0.43 \times 10^9$/l at baseline (*Sfikakis et al., 2005*). However, there were no effect of a single dose of RTX ($375$ mg/m$^2$) on the CD4 and CD8 cell counts as well as the percentage of the different CD4 cell subsets including regulatory T cells at three and 24 months in renal transplant recipients concomitantly treated with tacrolimus and mycophenolate mofetil during the first six months (*Kamburova et al., 2014*).

### Rituximab failed to normalise the CD4/CD8 ratio at 24 months

RTX induction and maintenance failed to normalise the CD4/CD8 ratio in GPA patients with an inverted ratio at baseline, although CD4 cell counts recovered. Being 10 years older at baseline doubled the risk of inversion of the CD4/CD8 ratio at 24 months. This suggests a relative increase in CD8+ lymphocytes in older GPA patients receiving RTX. Still, it remains unclear how absolute CD4 cell count and ratio related to changes in disease activity, subsets of CD4 and CD8 cells or the combination of toxic drug effects.

GPA patients with inverted ratio after 24 months of RTX maintenance seemed to have more disease activity prior to RTX, more kidney involvement and higher inflammation parameters, in accordance to a previous report from *Iking-Konert et al. (2008)*. They also had lower serum Ig during the course of our study, suggesting a more profound B cell-depleting effect.

### Limitations of the study

Our results should be interpreted with caution given the small size of the study cohort and the inherent risk of selection bias. Most of our GPA patients received RTX for refractory and relapsing disease, indicating a selected group who had received high cumulative dose of CYC and prolonged corticosteroids exposure prior to RTX. In addition, we only followed CD4 and CD8 counts in patients, and did not study important T cell subsets such as regulatory T cells.

### CONCLUSIONS

Our study suggests that the early decrease in CD4 after induction with RTX seemed dependent of the baseline CD4 count. Overall, CD4 cell count and CD4/CD8 ratio in GPA patients increased over time during RTX with no discernible impact of other immunosuppressive therapy. Increase in CD4 was not always followed by an increase in CD4/CD8 ratio, especially for a subgroup of older patients whom CD4/CD8 ratio remained inverted. Inverted CD4/CD8 ratio associated with lower Ig levels during maintenance with RTX.

Our study suggest that GPA patients with inverted CD4/CD8 ratio seemed to have a more profound B cell-depleting effect of RTX and a relative increase in CD8+ lymphocytes. CD4/CD8 ratio could be an important marker of a patient's net status of

immunodeficiency during RTX, since inverted CD4/CD8 ratio is a common surrogate marker of immunosenescence of impaired responses to vaccination and infections due to the loss of repertoire diversity (*Blackman & Woodland, 2011*).

### Funding
The authors received no funding for this work.

### Competing Interests
The authors declare that they have no competing interests.

### Author Contributions
- Emilio Besada analyzed the data, wrote the paper, prepared figures and/or tables, reviewed drafts of the paper.
- Johannes C. Nossent wrote the paper, reviewed drafts of the paper.

### Human Ethics
The following information was supplied relating to ethical approvals (i.e., approving body and any reference numbers):

Data are collected from the Vasculitis Registry in Northern Norway (NordNorsk vaskulitt register).

The study did not require formal ethical approval and conformed to the standards currently applied in Norway.

### Data Deposition
The raw data is available at Dataverse, Harvard University: http://dx.doi.org/10.7910/DVN/6U2KJ0.

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
