# Peer review of "CD4 cell count and CD4/CD8 ratio increase during rituximab maintenance in granulomatosis with polyangiitis patients"

_PeerJ, doi:10.7717/peerj.2487_

## Round 0.1 · original submission · Major Revisions

Your manuscript has been evaluated by two independent reviewers, who considered it potentially acceptable for publication in PeerJ.

However, one reviewer raised a number of criticisms that should be carefully considered. Please expand the Methods and Results sections following his/her suggestions.

Reviewer 1 ·

Basic reporting

no comment

Experimental design

Methods regarding the CD4 and CD8 determination should be discussed.

Validity of the findings

No comment

Additional comments

Dear Authors

I have read with interest your paper. Although the research is overall well conducted and some of the results of potential interest, I do feel some changes are required before it could be accepted.

MAJOR ISSUES
- Methods regarding the CD4 determination are not specified.
- I think that the results discussed in the section “results” should be presented while several times you talk about findings without showing the actual data (“data not shown”). I feel it would be more correct modify the paper reporting these data; alternatively I would suggest not mentioning them in the section “results” and maybe talk about them in the discussion section if felt necessary. Please, consider modifying the paper accordingly.
- In the discussion it looks like that the behaviour in terms of low CD4 and CD4/CD8 ratio is independent on the fact that RTX is used as maintenance treatment, this does not seem clear to me from the results section; although I appreciate the number of patients in your cohort not receiving maintenance rituximab is very low, I think a section in the results should be dedicated to that.
- Figure 1 and paper text referring to it: it is not specified anywhere what the numbers are: mean? Median? Other? This need to be clarified in the method section and in the figure explanation. Due to the small numbers, in order to have an idea of the degree of reliability of the data, I would suggest to report range or interquartile range in the form of “error bars” in the graphs.

MINOR ISSUES
Line 119, ODPD and CYC: need to be defined somewhere also in the text before using these abbreviations.
Line 132: how was the low CD4 definition established? Please, clarify and report references.
Table 1: the use of the term “control” for the patients without inverted ratio is misleading.

·

Basic reporting

No comments

Experimental design

No comments

Validity of the findings

No comments

Additional comments

The article is interesting and well written. The discussion is well structured and consistent with the experimental data

---

## Round 0.2 · Minor Revisions

FIGURE 1 is still unclear. Please add Y and X labels in the graph, and also report error bars, as suggested by Reviewer 1. Figures must be self-explanatory.

---

## Round 0.3 · accepted · Accept

The revised Figure 1 is now acceptable.